# Radiomics Models to Predict Tumor Response and Pneumonitis in Non-Small Cell Lung Cancer Patients Treated with Immunotherapy

**DOI:** 10.3390/jcm14124330

**Published:** 2025-06-18

**Authors:** Monica Yadav, Wongi Woo, Young Kwang Chae, Jeeyeon Lee, Peter Haseok Kim, Seyoung Lee, Taegyu Um, Salie Lee, Maria Jose Aguilera Chuchuca, Trie Arni Djunadi, Liam Il-Young Chung, Jisang Yu, Nicolo Gennaro, Leeseul Kim, Myungwoo Nam, Youjin Oh, Sungmi Yoon, Zunairah Shah, Yuchan Kim, Ilene Hong, Jessica Jang, Grace Kang, Amy Cho, Soowon Lee, Timothy Hong, Cecilia Nam, Yury S. Velichko

**Affiliations:** 1Feinberg School of Medicine, Northwestern University, NMH/Arkes Family Pavilion Suite 800, 676 N Saint Clair, Chicago, IL 60611, USA; monicay142@gmail.com (M.Y.); seyoung.b.lee@gmail.com (S.L.); alfred20@naver.com (T.U.); mariajose.aguilera.md@gmail.com (M.J.A.C.); eureka10chung@yahoo.com (L.I.-Y.C.); nicolo.gennaro@northwestern.edu (N.G.); sungmiyoon1@gmail.com (S.Y.); willkim012@gmail.com (Y.K.); 2St. Joseph’s Medical Center, 1800 N California St, Stockton, CA 95204, USA; wongi.woo@commonspirit.org; 3Department of Breast Surgery, Kyungpook National University, Daegu 37224, Republic of Korea; j.lee@knu.ac.kr; 4Department of Biochemisty, The University of Texas at Austin, Austin, TX 78712, USA; hkim20919@gmail.com; 5Division of Pulmonary Diseases and Critical Care Medicine, School of Medicine at UCI, Irvine, CA 92617, USA; md.s.lee.511@gmail.com; 6Richmond University Medical Center, Staten Island, NY 10310, USA; tdjunadi@rumcsi.org; 7Dignity Health—St. Rose Dominican Hospital, Henderson, NV 89113, USA; jisangyu.md@gmail.com; 8Ascension Saint Francis Hospital, Evanston, IL 60202, USA; leeseul.kim@uchicagomedicine.org; 9Lincoln Medical Centre, Bronx, NY 10451, USA; 10John H. Stroger, Jr. Hospital of Cook County, Chicago, IL 60612, USA; oujin0521@gmail.com; 11Roswell Park Comprehensive Cancer Center, Buffalo, NY 14203, USA; zunairahs.zs@gmail.com (Z.S.); cecilianam2025@u.northwestern.edu (C.N.); 12Evanston Campus, Northwestern University, Evanston, IL 60208, USA; ilenehong2024@u.northwestern.edu (I.H.); jessicajjang11@gmail.com (J.J.); gracekang2025@u.northwestern.edu (G.K.); yubeencho2025@u.northwestern.edu (A.C.); timjisukhong@gmail.com (T.H.); 13Department of Global Disease Epidemiology and Control, Johns Hopkins Bloomberg School of Public Health, Baltimore, MD 21205, USA; slee623@jh.edu

**Keywords:** radiomics, immunotherapy, lung cancer, pneumonitis

## Abstract

**Background:** Checkpoint inhibitor-associated pneumonitis (CIP) after immunotherapy has become a challenging issue in non-small cell lung cancer (NSCLC) patients. This study leverages artificial intelligence (AI) algorithms to analyze radiomic features, aiming to predict the occurrence of CIP, as well as tumor response. **Methods:** This study analyzed data from 159 stage III-IV NSCLC patients undergoing immunotherapy. The patients were categorized into pneumonitis and non-pneumonitis groups, and 3D radiomic features from both tumors and surrounding regions were extracted using LIFEx software. To address scanner-associated variations, a linear mixed-effect radiomics harmonization model was applied. A random forest algorithm was then used to develop models predicting CIP occurrence and tumor responses based on the pre-treatment CT radiomics. The accuracy was evaluated using the area under the curve (AUC). **Results:** A total of 159 patients were analyzed, of which only 31 experienced CIP. Most had grade 1 (17/31, 54.8%) or 2 (12/31, 38.7%) pneumonitis; only two (6.5%) patients had grade 3. Patients who developed pneumonitis were more likely to be male (64.5% vs. 38.3%, *p* = 0.014), had less adenocarcinoma histology (54.8% vs. 78.9%, *p* = 0.032), and exhibited a higher tumor mutational burden (57.1% vs. 24.5%, *p* = 0.047). Radiomics analysis reported predictability for CIP with an AUC of 0.60 (95% CI 0.55–0.66). The five-year overall and progression-free survival rates were 24.7% (95% CI 15.2–35.5%) and 9.7% (95% CI 4.4–17.4%), respectively. The radiomics features also exhibited AUCs of 0.63 (95% CI 0.59–0.67) in irRECIST and 0.66 (95% CI 0.61–0.70) in RECIST 1.1 in terms of tumor responses to immunotherapy. **Conclusions:** This study provides insights into the potential role of radiomic models in predicting CIP and tumor responses from pre-treatment CT images of NSCLC patients treated with immunotherapy.

## 1. Introduction

The American Cancer Society’s projections for 2024 estimate 2,001,140 new cancer cases and 611,720 cancer-related deaths in the U.S., utilizing the latest data from the National Center for Health Statistics and central cancer registries [1]. Lung cancer remains the leading cause of cancer death globally, posing significant public health challenges due to its late diagnosis—often at stage III or IV for over 70% of patients—and resulting in a low five-year survival rate of 20–30% [2]. This emphasizes the critical need for early detection and intervention strategies to improve outcomes among lung cancer patients.

Immunotherapy has revolutionized the treatment of lung cancer, offering new hope for patients with advanced stages [3,4]. It works by empowering the immune system to recognize and attack cancer cells more effectively. A key player in this therapy is the protein programmed death-ligand 1 (PD-L1), which some cancer cells express to evade the immune system. Certain immunotherapies, known as checkpoint inhibitors, target the interaction between PD-L1 on cancer cells and its receptor programmed cell death protein 1 (PD-1) on immune cells. By blocking this interaction, these drugs prevent cancer cells from hiding, thereby allowing the immune system to detect and destroy them.

Although high PD-L1 expressions are generally associated with a better response to PD-1/PD-L1 inhibitors, there are some patients in the gray zone: even patients with low or no PD-L1 expression respond to immunotherapy. In addition, some patients experience life-threatening complications related to immunotherapy, such as checkpoint inhibitor-related pneumonitis (CIP). The number of patients who develop CIP is increasing because of the expanding indications of immunotherapy in NSCLC [5,6]. Thus, the precise prediction of immunotherapy response to cancer shrinkage and the prediction of adverse events in NSCLC is crucial for personalized immunotherapy. These limitations underscore the need for complementary biomarkers and more comprehensive approaches to predict immunotherapy outcomes more accurately.

Radiomics is a technique that extracts quantitative features from medical images using data-characterization algorithms. These radiomic features can be used to identify tissue characteristics and radiologic phenotyping that is not observable by clinicians [7,8,9]. Following their extraction, radiomic features can be used in a statistical or deep learning paradigm to develop models predictive of various clinical outcomes.

This study investigates the application of pretreatment computed tomography (CT)-based radiomics and leverages artificial intelligence (AI) algorithms and the use of harmonization models to analyze radiomic features from chest CT scans, aiming to predict the tumor response and the occurrence of CIP in patients with NSCLC.

## 2. Methods

### 2.1. Study Cohort

A retrospective review of electronic medical records was conducted on 612 NSCLC patients who underwent immunotherapy at Northwestern Memorial Hospital from 1 January 2013 to 31 December 2022. The inclusion criteria for this study were as follows: (1) clinically unresectable stages III–IV; (2) immunotherapy-naïve patients; (3) having at least one measurable intrathoracic lesion that was large enough (≥10 mm) to extract radiomic textures. The exclusion criteria were as follows: (1) patients with advanced stages without intrathoracic lesions; (2) patients who did not undergo a chest CT before and after first-line immunotherapy; (3) patients without contrast-enhanced chest CT scans; (4) images with poor quality inadequate for radiomics analysis; or (5) incomplete clinical data regarding post-treatment status.

### 2.2. Clinical Variables of Interest

The baseline demographic data included age, sex, body-mass index, Eastern Cooperative Oncology Group performance status (ECOG PS), smoking history, tumor stage, and tumor histology. Further variables related to immunotherapy were also investigated, including the immunotherapy regimens used, PD-L1 tumor staining, the neutrophil–lymphocyte ratio, the tumor mutational burden value, microsatellite instability, and the driver gene mutation profile.

## 3. Radiomics Features

### 3.1. Image Extraction

Northwestern Memorial Hospital granted permission to acquire contrast-enhanced CT chest scans for quantitative analysis. All patients underwent these scans at baseline and immediately following two to three cycles (6–8 weeks) of immunotherapy. Prior to inclusion in the study, these images were de-identified to protect patient privacy. The de-identified images were then provided to the researchers for detailed analysis. Cancer-related lesions were identified by experienced radiologists through careful review of the original CT images and corresponding radiology reports. Only intrathoracic lesions involving the lung parenchyma and peritumoral space were included in this study (Figure 1).

### 3.2. Image Segmentation and Feature Extraction

The solid tumor and its surrounding peri-tumoral region were manually segmented by three physicians (M.Y., J.L., and Y.S.V.) using a free multi-platform tool (LIFEx software, Version 7.4, IMIV/CEA, Orsay, France) [10]. Measurable target lesions in the lung, with a longest axial diameter greater than 10 mm, were selected for analysis. A 10 mm thick peritumoral region was automatically created using LIFEx software. To improve the accuracy of radiomics analysis, the tumor and peritumoral region from contrast-enhanced CT scans were segmented using thresholds of −50 to 300 HU for the tumor and −1000 to −50 HU for the peritumoral space. Tumor lesions trapped within pleural effusion were excluded.

A total of 1518 3D radiomic features were extracted from the tumor and peritumoral region, including first-order statistics, shape-based metrics, gray level co-occurrence matrix (GLCM), gray level run length matrix (GLRLM), gray level size zone matrix (GLSZM), neighboring gray tone difference matrix (NGTDM), and gray level dependence matrix (GLDM). To address scanner-related variability, a linear mixed-effects harmonization model was employed using three spherical tissue samples of spleen and normal lung parenchyma per patient, with volume of 1–2 cm^3^, 3–5 cm^3^, and 5–10 cm^3^, respectively [11] (Figure 2).

### 3.3. Prediction Model Development

The random forest algorithm was employed to develop a classification model to differentiate between responders and non-responders. A harmonization model was integrated to account for the scanner-related differences using spleen and normal lung signals. The dataset was divided into a training set (75%) and a test set (25%). Bootstrapping with 1000 iterations was performed to estimate the model’s performance by calculating the median and 95% confidence interval (CI) estimate. The accuracy of the model’s predictions was evaluated by creating a confusion matrix. The model’s performance was assessed by calculating the sensitivity, specificity, positive predictive values (PPV), negative predictive values (NPV), and area under the receiver operating characteristic curve (AUC) for response prediction. The same algorithm, dataset division, and bootstrapping were used to differentiate between CIP and non-CIP. The performance was also evaluated with the same values that were used for response prediction.

## 4. Outcomes of Interest

Progression-free survival was defined as the time from initial immunotherapy until disease progression or intolerable treatment toxicity. Overall survival was calculated from the date of immunotherapy to death from any cause or the date of the last follow-up. Regarding the clinical response to immunotherapy, it was evaluated based on the Response Evaluation Criteria in Solid Tumors 1.1 (RECIST 1.1) and the RECIST-based immune criteria (irRECIST).

The primary outcome was the prediction of checkpoint inhibitor-related pneumonitis (CIP). CIP was graded according to a standard scale of severity, the Common Terminology Criteria for Adverse Events Version 5.0 (CTCAE V5).

The secondary outcome was the evaluation of the treatment response of intrathoracic lesions based on two criteria: RECIST 1.1 and irRECIST. This study focused on predicting durable responder group, which consisted of complete response (CR), partial response (PR), or stable disease (SD) for a minimum duration of 24 weeks. Non-responder group comprised patients who experienced progressive disease (PD) or only achieved SD with a duration shorter than 24 weeks.

## 5. Statistical Analysis

Continuous variables were presented as the medians and interquartile ranges due to their non-normal distribution, and the Mann–Whitney U test was applied to compare differences among them. Fisher’s exact test was used to compare categorical variables. All statistical analyses were performed using R version 4.0.4 (R Core Team, R Foundation for Statistical Computing, Vienna, Austria). Differences with a two-tailed *p* value < 0.05 were considered statistically significant.

## 6. Results

### 6.1. Study Population and Characteristics

A total of 159 patients were included in this retrospective study. Table 1 describes the patients’ characteristics at the time of immunotherapy initiation. Most patients had good/fair functional status, with an ECOG score of 0–1 (139/159, 87.4%), and adenocarcinoma (118/159, 74.2%) was the most common histologic type. The majority of patients had a stage IV diagnosis (91.2%), and PD-L1 staining was positive in 63.9% (85/133) of those tested. In terms of treatments, over half of the patients (96/159) received a single immune checkpoint inhibitor (ICI), while the others underwent combination chemotherapy. In terms of ICI regimens, pembrolizumab was administered to about half the patients (86/159, 54.1%) followed by nivolumab (24/159, 15.1%), atezolizumab (22/159, 13.8%), and nivolumab with ipilimumab (14/159, 8.8%).

After the next-generation sequencing (NGS) of tumor lesions in 79 patients, high TMB over 10 mutation/Mb was observed in a third of the patients (20/63, 31.7%). The most common driver gene mutations observed in the NGS were as follows: TP53 (91/125, 72.8%), TTF1 (85/153, 55.6%), KRAS (44/150, 29.3%), EGFR (33/152, 21.7%), BRAF (21/104, 20.2%), and CDKN2A (21/142, 14.8%), among others (Table 1).

### 6.2. Pneumonitis Prediction via Radiomics Analysis

Pneumonitis occurred in 19.4% (31/159) of the patients, and classification between radiotherapy- and immunotherapy-induced pneumonitis was based on clinical judgement in the medical records. In terms of the severity of pneumonitis (Table 2), most had a mild degree of pneumonitis (grade 1: 17/31(54.8%); grade 2: 12/31 (38.7%)) and only 6.5% (2/31) had severe pneumonitis, classified as grade 3. No mortality related to CIP was observed.

Figure 3 exhibits an accuracy performance histogram for each radiomics feature in the random forest algorithms. Compared to clinical variables, such as the neutrophil–lymphocyte ratio (NLR) and platelet counts, the radiomics features did not provide convincing accuracy, most likely due to the small group size. The neighboring gray tone difference matrix (NGTDM) and gray level co-occurrence matrix (GLCM) presented relatively moderate prognostic values.

In general, the predictability of pneumonitis after immunotherapy was suboptimal (Table 4) based on the radiomics variables. The AUC in regard to all pneumonitis types was 0.60 (95% CI 0.55–0.66), while the accuracy for immunotherapy-related pneumonitis had an AUC of 0.59 (95% CI 0.56–0.66).

## 7. Clinical Outcomes

During a median of 14.5 months of follow-up, about half of the patients died (87/159, 54.7%). The overall and progression-free survival rates are presented in Figure 4. The three- and five-year OS rates were 43.5% (95% CI 34.3–52.4%) and 24.7% (95% CI 15.2–35.5%), respectively. The three- and five-year PFS rates were 21.0% (95% CI 14.0–28.8%) and 9.7% (95% CI 4.4–17.4%), respectively. Regarding PD-L1 expression, PFS and OS exhibited significant differences according to their levels. In terms of tumor response to treatment based on irRECIST criteria, 26.4% (42/159) exhibited progressive disease after ICI therapy (Table 1).

## 8. Responses to Immunotherapy

Tumor responses to immunotherapy were evaluated by two criteria: RECIST1.1 and irRECIST. Table 3 shows the clinical characteristics based on tumor response. Durable responders tended to have a higher tumor mutational burden than the non-responders (*p* = 0.003). Other factors were insignificant, including demographic variables.

Table 4 shows the predictability of the radiomics features in terms of tumor response. Tumor response was categorized into three variables: PD, SD, and PR + CR. The overall accuracy of the radiomics analysis showed an AUC of 0.63 (95% CI 0.59–0.67) in irRECIST and that of 0.66 (95% CI 0.61–0.70) in RECIST 1.1.

## 9. Discussion

This study demonstrated the results of radiomics analysis from pre-treatment CT images to predict immunotherapy-related pneumonitis and tumor responses. Without integrating clinical variables, the radiomics features themselves made meaningful contributions to determining who was vulnerable to this immunotherapy-associated adverse outcome. Furthermore, these features were applicable to predict tumor responses from immunotherapy. This study demonstrated how useful this artificial intelligence algorithm is to predict clinical outcomes, but it also exhibited several significant challenges in terms of reproducibility in real-world settings.

The pathogenesis of immunotherapy-related pneumonitis is not still clear, and there are several plausible suggestions [12]. One possibility is an increased T-cell reaction against cross-antigens expressed in normal and tumor cells [13]. Another option is increased levels of preexisting and emerging autoantibodies, which are triggered by immunotherapy [14]. A third possibility is unbalanced levels of pro- and anti- inflammatory cytokines, such as C-reactive protein (CRP), IL-6, and IL-17. Change in these variables have been observed in several studies [15,16]. Due to the limited numbers of trials and validation results, further studies are necessary to determine the underlying scientific reasoning.

Differentiating the etiology of pneumonitis from RT or ICI is still the subject of ongoing debate. Traditionally, the timing of onset and the distribution area of RT were commonly applied to discern these two etiologies. However, even multidisciplinary discussions and reviews of medical records have not reached a high level of agreement among clinicians [17]. Therefore, radiomics and machine learning algorithms have been implemented, with the benefits of a comprehensive review of imaging studies [12,17]. Our study also attempted to differentiate between two etiologies, but it was not sufficient due to the small sample size, which has been commonly observed in other studies. As clinicians obtain more information on the underlying mechanisms and phenotypical manifestation of these two types of pneumonitis, the predictability of radiomics is expected to improve significantly in the future.

Tumor response prediction is also another issue for patients. Given molecular biomarkers and targeted therapies, there has been significant growth in the field of precision oncology. However, more insights are needed to understand the diverse responses to immunotherapy other than PD-L1 expression. Tumor heterogeneity and the tumor microenvironment have recently attracted significant attention from clinical societies. Several studies have explored the role of radiomics in terms of tumor responses [9,18]. However, complex clinical situations should be considered. Trebeschi et al. reported better performance in NSCLC patients than in melanoma patients due to their more homogeneous first-line treatments [9]. Therefore, combining clinical and radiomics signatures will be necessary to apply in real-world situations.

Compared to other studies, our study did not present superior predictions of immunotherapy-related pneumonitis; previous studies with smaller sample sizes demonstrated AUCs value of around 0.80–0.90 [17,19,20]. This represents challenges in the application of radiomics algorithms to actual clinical fields. There are significant limitations causing this. The segmentation of lesions by investigators or software is challenging and its repeatability remains a main cause of suboptimal outcomes. Also, this study only included intrathoracic lesions, which may not reflect real-world clinical scenarios of multiple extrathoracic metastasis. Furthermore, several key clinical factors (genomic alteration, pathology, prior chemotherapy, and molecular findings) were not integrated into our models. Therefore, it is necessary to improve the performance of the prediction system. To apply these techniques universally, further validation with prospective trials and external dataset is necessary, and definitions of radiologic features should be standardized. As this study did not have enough participants, it is also difficult to generalize the findings. Therefore, the application of these findings to real-world clinical practice may not be appropriate at this time.

## 10. Conclusions

In a cohort of stage III or IV NSCLC patients who were treated with immunotherapy, radiomics analysis exhibited acceptable rates in predicting clinical outcomes, such as pneumonitis and tumor responses. Though the predictive and prognostic implications of radiomics models for these outcomes are limited due to the complexity of tumor biology, their performance could be improved by incorporating other relevant clinical and molecular features. By doing so, the precise application of immunotherapy and selective monitoring during immunotherapy would lead to improvements in the efficacy and outcomes of radiomics analysis.

## Figures and Tables

**Figure 1 jcm-14-04330-f001:**
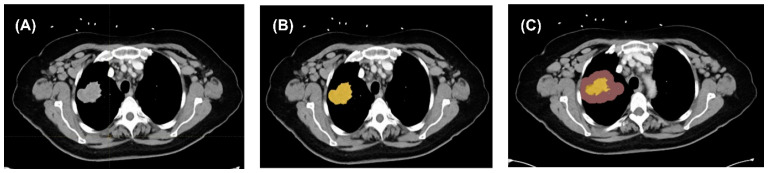
Image segmentation process. (**A**) Contrast-enhanced images of region of interest. (**B**) Tumoral segmentation. (**C**) Peritumoral and tumoral segmentation.

**Figure 2 jcm-14-04330-f002:**
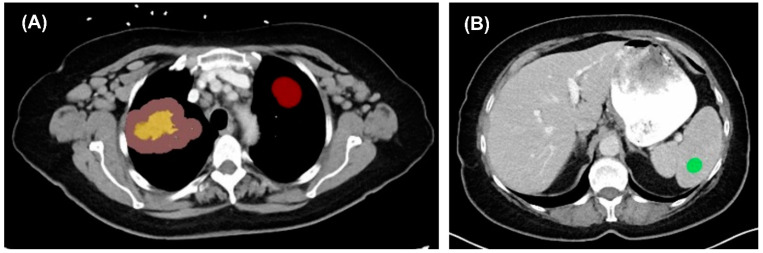
Harmonization process for radiomics feature analysis. (**A**) Red area represents harmonization with normal lung area. (**B**) Green area represents harmonization with normal spleen area.

**Figure 3 jcm-14-04330-f003:**
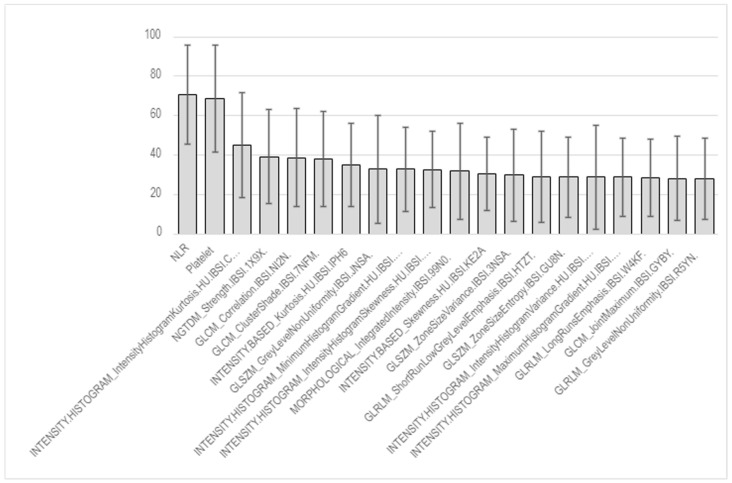
Accuracy performance histogram for each radiomics feature in random forest algorithms.

**Figure 4 jcm-14-04330-f004:**
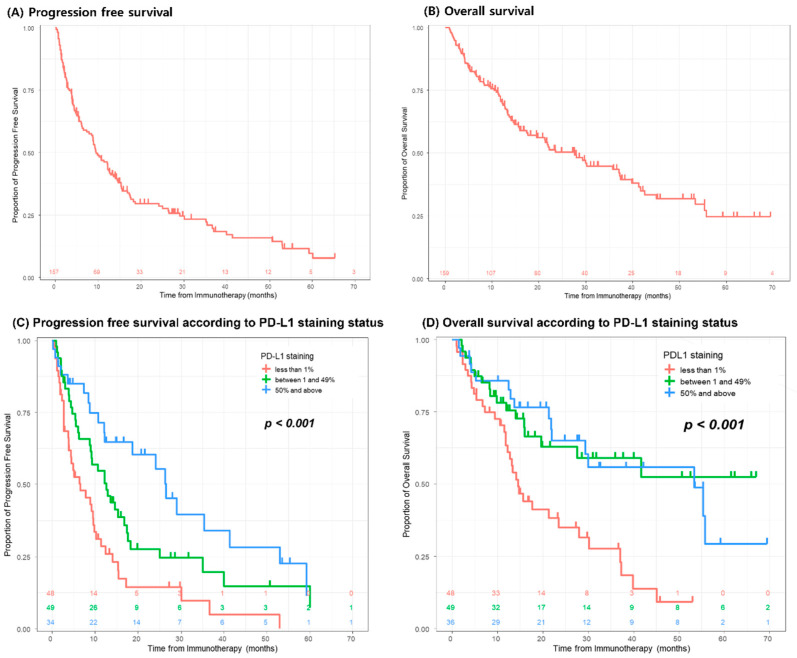
Clinical outcomes of the entire cohort and based on PD-L1 expression levels. PD-L1, programmed death-ligand 1.

**Table 1 jcm-14-04330-t001:** Clinical characteristics of patients with or without pneumonitis development.

Clinical Variables		Overall	Pneumonitis		
			No	Yes	*p*-Value
		N = 159	N = 128	N = 31	
Age, years		66.0 [59.0, 74.0]	65.5 [59.0, 74.0]	68.0 [61.0, 74.5]	0.303
Body mass index, kg/m^2^		25.6 [22.3, 30.3]	25.2 [22.0, 30.5]	27.2 [23.1, 29.9]	0.435
Sex	Female	90 (56.6)	79 (61.7)	11 (35.5)	0.014
	Male	69 (43.4)	49 (38.3)	20 (64.5)	
Smoking history	Never smoked	33 (20.8)	29 (22.7)	4 (12.9)	0.428
	Former smoker	111 (69.8)	88 (68.8)	23 (74.2)	
	Current smoker	15 (9.4)	11 (8.6)	4 (12.9)	
ECOG status	ECOG 0	81 (50.9)	67 (52.3)	14 (45.2)	0.476
	ECOG 1	58 (36.5)	44 (34.4)	14 (45.2)	
	ECOG 2	17 (10.7)	15 (11.7)	2 (6.5)	
	ECOG 3	3 (1.9)	2 (1.6)	1 (3.2)	
Histology	Adenocarcinoma	118 (74.2)	101 (78.9)	17 (54.8)	0.032
	Squamous cell	29 (18.2)	18 (14.1)	11 (35.5)	
	Adenosquamous	2 (1.3)	2 (1.6)	0 (0.0)	
	Large cell	1 (0.6)	1 (0.8)	0 (0.0)	
	Others	9 (5.7)	6 (4.7)	3 (9.7)	
Staging, 8th TNM	III	14 (8.8)	10 (7.8)	4 (12.9)	0.477
	IV	145 (91.2)	118 (92.2)	27 (87.1)	
PD-L1—tumor staining	<1%	48 (36.1)	39 (37.1)	9 (32.1)	0.531
	1–49%	49 (36.8)	36 (34.3)	13 (46.4)	
	≥50%	36 (27.1)	30 (28.6)	6 (21.4)	
Baseline laboratory findings					
Platelet counts		280 [220, 351]	288 [221, 353]	263 [200, 343]	0.242
Lymphocyte proportion, %		16.0 [10.3, 23.5]	17.0 [11.0, 24.3]	13.2 [9.0, 20.0]	0.086
Neutrophil proportion, %		71.0 [63.0, 79.0]	70.0 [61.8, 78.0]	71.0 [64.0, 81.0]	0.486
NLR		4.41 [2.67, 7.41]	4.33 [2.54, 6.75]	5.26 [3.07, 9.06]	0.142
Treatment	ICI only	96 (60.4)	77 (60.2)	19 (61.3)	1
	ICI with CTx	63 (39.6)	51 (39.8)	12 (38.7)	
ICI Regimens	Pembrolizumab	86 (54.1)	71 (55.5)	15 (48.4)	0.267
	Durvalumab	10 (6.3)	5 (3.9)	5 (16.1)	
	Nivolumab only	24 (15.1)	20 (15.6)	4 (12.9)	
	Nivolumab/ipilimumab	14 (8.8)	10 (7.8)	4 (12.9)	
	Atezolizumab	22 (13.8)	19 (14.8)	3 (9.7)	
	Ramucirumab	2 (1.3)	2 (1.6)	0 (0.0)	
	Cemiplimab	1 (0.6)	1 (0.8)	0 (0.0)	
TMB, mutations/Mb		8.0 [3.8, 11.7]	6.7 [3.2, 9.8]	10.8 [6.4, 19.7]	0.059
	TMB <10	43 (68.3)	37 (75.5)	6 (42.9)	0.047
	TMB ≥ 10	20 (31.7)	12 (24.5)	8 (57.1)	
Microsatellite instability	Negative	67 (84.8)	54 (85.7)	13 (81.2)	0.701
	Positive	12 (15.2)	9 (14.3)	3 (18.8)	
Veristrat	1. Good	38 (66.7)	30 (65.2)	8 (72.7)	0.735
	2. Poor	19 (33.3)	16 (34.8)	3 (27.3)	
Identified gene mutations					
ALK		11/133 (8.3)	8/108 (7.4)	3/25 (12.0)	0.432
ARID1A		15/154 (9.7)	14/123 (11.4)	1/31 (3.2)	0.307
ATM		16/154 (10.4)	14/123 (11.4)	2/31 (6.5)	0.529
BRAF		21/104 (20.2)	19/84 (22.6)	2/20 (10.0)	0.352
BRCA1		7/154 (4.5)	3/123 (2.4)	4/31 (12.9)	0.031
BRCA2		10/154 (6.5)	7/123 (5.7)	3/31 (9.7)	0.422
CDKN2A		21/142 (14.8)	17/113 (15.0)	4/29 (13.8)	1
CDKN2B		8/154 (5.2)	7/123 (5.7)	1/31 (3.2)	1
EGFR		33/152 (21.7)	25/122 (20.5)	8/30 (26.7)	0.465
ERBB2		9/157 (5.7)	9/126 (7.1)	0/31 (0.0)	0.207
KRAS		44/150 (29.3)	41/121 (33.9)	3/29 (10.3)	0.012
MET		14/158 (8.9)	10/127 (7.9)	4/31 (12.9)	0.478
NF1		13/155 (8.4)	11/124 (8.9)	2/31 (6.5)	1
NOTCH1		11/157 (7.0)	7/126 (5.6)	4/31 (12.9)	0.229
PIK3CA		17/158 (10.8)	12/127 (9.4)	5/31 (16.1)	0.331
PTEN		10/158 (6.3)	7/127 (5.5)	3/31 (9.7)	0.413
RB1		7/154 (4.5)	4/123 (3.3)	3/31 (9.7)	0.146
ROS1		10/108 (9.3)	8/86 (9.3)	2/22 (9.1)	1
STK11		13/158 (8.2)	13/127 (10.2)	0/31 (0.0)	0.074
TP53		91/125 (72.8)	72/99 (72.7)	19/26 (73.1)	1
TTF1		85/153 (55.6)	70/123 (56.9)	15/30 (50.0)	0.542
Follow-up duration, months		14.5 [6.9, 30.1]	14.9 [7.4, 30.2]	12.7 [5.9, 28.9]	0.419
Mortality		87/159 (54.7)	68/128 (53.1)	19/31 (61.3)	0.432
Tumor response by irRECIST	CR	1 (0.6)	1 (0.8)	0 (0.0)	1
	PD	42 (26.4)	34 (26.6)	8 (25.8)	
	PR	51 (32.1)	41 (32.0)	10 (32.3)	
	SD	65 (40.9)	52 (40.6)	13 (41.9)	
Tumor response by RECIST1.1	CR	1 (0.6)	1 (0.8)	0 (0.0)	0.653
	PD	56 (35.2)	47 (36.7)	9 (29.0)	
	PR	40 (25.2)	30 (23.4)	10 (32.3)	
	SD	62 (39.0)	50 (39.1)	12 (38.7)	

Data are presented as n (%), n/N (%), or median [interquartile range]. CTx, chemotherapy; CR, complete response; ICI, immune checkpoint inhibitor; ECOG, Eastern Cooperative Oncology Group performance status; NLR, neutrophil–lymphocyte ratio; PD-L1, programmed death-ligand 1; PD, progressive disease; PR, partial response; SD, stable disease; TMB, tumor mutational burden.

**Table 2 jcm-14-04330-t002:** Pneumonitis classification according to severity and possible causes.

Clinical Variables		Number of Patients
Severity by CTCAE V5		
	Grade 1	17/31 (54.8)
	Grade 2	12/31 (38.7)
	Grade 3	2/31 (6.5)
Types by causes		
	ICI-related	18/29 (62.1)
	Mixed	1/29 (3.4)
	Radiation-related	10/29 (34.5)

Data are presented as n/N (%). CTCAE V5, Common Terminology Criteria for Adverse Events Version 5.0; ICI, immune checkpoint inhibitor.

**Table 3 jcm-14-04330-t003:** Clinical characteristics of patients according to tumor response by irRECIST.

Clinical Variables		Overall	Tumor Response		
			Durable Responder	Non-Responder	*p*-Value
		N = 159	N = 117	N = 42	
Age, years		66.0 [59.0, 74.0]	67.0 [59.0, 76.0]	64.5 [59.0, 71.7]	0.174
Body mass index, kg/m^2^		25.6 [22.3, 30.3]	25.3 [22.0, 30.4]	25.8 [22.9, 29.8]	0.467
Sex	Female	90 (56.6)	66 (56.4)	24 (57.1)	1
	Male	69 (43.4)	51 (43.6)	18 (42.9)	
Smoking history	Never smoked	33 (20.8)	23 (19.7)	10 (23.8)	0.808
	Former smoker	111 (69.8)	82 (70.1)	29 (69.0)	
	Current smoker	15 (9.4)	12 (10.3)	3 (7.1)	
ECOG status	ECOG 0	81 (50.9)	55 (47.0)	26 (61.9)	0.086
	ECOG 1	58 (36.5)	49 (41.9)	9 (21.4)	
	ECOG 2	17 (10.7)	11 (9.4)	6 (14.3)	
	ECOG 3	3 (1.9)	2 (1.7)	1 (2.4)	
Histology	Adenocarcinoma	118 (74.2)	86 (73.5)	32 (76.2)	0.722
	Squamous cell	29 (18.2)	23 (19.7)	6 (14.3)	
	Adenosquamous	2 (1.3)	1 (0.9)	1 (2.4)	
	Large cell	1 (0.6)	1 (0.9)	0 (0.0)	
	Others	9 (5.7)	6 (5.1)	3 (7.1)	
Staging, 8th TNM	III	14 (8.8)	12 (10.3)	2 (4.8)	0.358
	IV	145 (91.2)	105 (89.7)	40 (95.2)	
PD-L1—Tumor staining	<1%	48/133 (36.1)	31/100 (31.0)	17/33 (51.5)	0.108
	1–49%	49/133 (36.8)	39/100 (39.0)	10/33 (30.3)	
	≥50%	36/133 (27.1)	30/100 (30.0)	6/33 (18.2)	
Baseline laboratory findings					
Platelet counts		280 [220, 351]	289 [222, 350]	267 [210, 364]	0.546
Lymphocyte proportion, %		16.0 [10.3, 23.5]	16.0 [10.6, 24.0]	17.0 [10.3, 23.0]	0.778
Neutrophil proportion, %		71.0 [63.0, 79.0]	70.0 [63.0, 79.0]	71.0 [63.2, 75.8]	0.973
NLR		4.41 [2.67, 7.41]	4.39 [2.67, 7.36]	4.49 [2.79, 7.98]	0.862
Treatment	ICI only	96 (60.4)	66 (56.4)	30 (71.4)	0.132
	ICI with CTx	63 (39.6)	51 (43.6)	12 (28.6)	
ICI regimens	Pembrolizumab	86 (54.1)	65 (55.6)	21 (50.0)	0.222
	Durvalumab	10 (6.3)	10 (8.5)	0 (0.0)	
	Nivolumab only	24 (15.1)	16 (13.7)	8 (19.0)	
	Nivolumab/ipilimumab	14 (8.8)	10 (8.5)	4 (9.5)	
	Atezolizumab	22 (13.8)	13 (11.1)	9 (21.4)	
	Ramucirumab	2 (1.3)	2 (1.7)	0 (0.0)	
	Cemiplimab	1 (0.6)	1 (0.9)	0 (0.0)	
TMB, bp/Mb		8.00 [3.75, 11.72]	9.15 [4.15, 16.45]	4.20 [2.11, 7.30]	0.003
	TMB <10	43/63 (68.3)	28/48 (58.3)	15/15 (100.0)	0.001
	TMB ≥ 10	20/63 (31.7)	20/48 (41.7)	0/15 (0.0)	
Microsatellite instability	Negative	67 (84.8)	50 (82.0)	17 (94.4)	0.278
	Positive	12 (15.2)	11 (18.0)	1 (5.6)	
Veristrat	(1) Good	38 (66.7)	31 (64.6)	7 (77.8)	0.703
	(2) Poor	19 (33.3)	17 (35.4)	2 (22.2)	
Identified gene mutations					
ALK		11/133 (8.3)	8/101 (7.9)	3/32 (9.4)	0.725
ARID1A		15/154 (9.7)	10/112 (8.9)	5/42 (11.9)	0.555
ATM		16/154 (10.4)	11/112 (9.8)	5/42 (11.9)	0.768
BRAF		21/104 (20.2)	14/84 (16.7)	7/20 (35.0)	0.117
BRCA1		7/154 (4.5)	5/112 (4.5)	2/42 (4.8)	1
BRCA2		10/154 (6.5)	7/112 (6.2)	3/42 (7.1)	1
CDKN2A		21/142 (14.8)	14/104 (13.5)	7/38 (18.4)	0.438
CDKN2B		8/154 (5.2)	5/112 (4.5)	3/42 (7.1)	0.684
EGFR		33/152 (21.7)	26/115 (22.6)	7/37 (18.9)	0.819
ERBB2		9/157 (5.7)	7/116 (6.0)	2/41 (4.9)	1
KRAS		44/150 (29.3)	32/113 (28.3)	12/37 (32.4)	0.679
MET		14/158 (8.9)	9/116 (7.8)	5/42 (11.9)	0.526
NF1		13/155 (8.4)	11/113 (9.7)	2/42 (4.8)	0.516
NOTCH1		11/157 (7.0)	9/115 (7.8)	2/42 (4.8)	0.728
PIK3CA		17/158 (10.8)	11/116 (9.5)	6/42 (14.3)	0.393
PTEN		10/158 (6.3)	8/116 (6.9)	2/42 (4.8)	1
RB1		7/154 (4.5)	6/112 (5.4)	1/42 (2.4)	0.675
ROS1		10/108 (9.3)	7/85 (8.2)	3/23 (13.0)	0.441
STK11		13/158 (8.2)	8/116 (6.9)	5/42 (11.9)	0.333
TP53		91/125 (72.8)	70/96 (72.9)	21/29 (72.4)	1
TTF1		85/153 (55.6)	63/112 (56.2)	22/41 (53.7)	0.855
Follow-up duration, months		14.5 [6.9, 30.1]	16.8 [9.4, 36.6]	10.7 [4.3, 16.5]	0.001
Mortality		87/159 (54.7)	55/117 (47.0)	32/42 (76.2)	0.432
Pneumonitis	Yes	31/159 (19.5)	23/117 (19.7)	8/42 (19.0)	1
Grade 1		17/31 (54.8)	12/23 (52.2)	5/8 (62.5)	0.760
Grade 2		12/31 (38.7)	10/23 (43.5)	2/8 (25.0)	
Grade 3		2/31 (6.5)	1/23 (4.3)	1/8 (12.5)	

Data are presented as n (%), n/N (%), or median [interquartile range]. CTx, chemotherapy; CR, complete response; ICI, immune checkpoint inhibitor; ECOG, Eastern Cooperative Oncology Group performance status; NLR, neutrophil–lymphocyte ratio; PD-L1, programmed death-ligand 1; PD, progressive disease; PR, partial response; SD, stable disease; TMB, tumor mutational burden.

**Table 4 jcm-14-04330-t004:** Diagnostic performance of radiomics features in pneumonitis development and tumor response to immunotherapy.

	Pneumonitis			Tumor Response by irRECIST			Tumor Response by RECIST 1.1		
	All	ICI	RTx	PD	SD	PR + CR	PD	SD	PR + CR
AUC	0.60 (0.55–0.66)	0.59 (0.56–0.66)	0.53 (0.46–0.80)	0.63 (0.59–0.67)			0.66 (0.61–0.70)		
Sensitivity	0.97 (0.95–0.98)	0.98 (0.97–0.99)	1.00 (0.99–1.00)	0.49 (0.45–0.56)	0.64 (0.60–0.68)	0.45 (0.38–0.50)	0.57 (0.52–0.62)	0.55 (0.52–0.60)	0.34 (0.28–0.38)
Specificity	0.08 (0.05–0.14)	0.06 (0.03–0.11)	0.04 (0.00–0.07)	0.82 (0.79–0.85)	0.63 (0.59–0.67)	0.85 (0.83–0.87)	0.65 (0.63–0.69)	0.67 (0.63–0.72)	0.91 (0.89–0.93)
PPV	0.83 (0.82–0.83)	0.91 (0.90–0.91)	0.93 (0.92–0.93)	0.56 (0.51–0.61)	0.54 (0.50–0.57)	0.53 (0.49–0.59)	0.52 (0.48–0.54)	0.51 (0.48–0.55)	0.51 (0.46–0.56)
NPV	0.38 (0.29–0.50)	0.27 (0.21–0.40)	0.14 (0.00–0.67)	0.78 (0.76–0.80)	0.72 (0.70–0.75)	0.79 (0.77–0.81)	0.71 (0.68–0.73)	0.71 (0.69–0.73)	0.82 (0.81–0.83)
Balanced Accuracy	0.52 (0.51–0.55)	0.53 (0.51–0.54)	0.51 (0.50–0.53)	0.65 (0.63–0.69)	0.64 (0.61–0.67)	0.65 (0.61–0.68)	0.62 (0.59–0.64)	0.61 (0.59–0.64)	0.62 (0.59–0.64)

AUC, area under the curve; CR, complete response; ICI, immune checkpoint inhibitor; NPV, negative predictive value; PPV, positive predictive value; PD, progression disease; PR, partial response; RTx, radiotherapy.

## Data Availability

The data underlying this article will be shared by the corresponding author upon reasonable request.

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
