# Peer review of "Radiomics Models to Predict Tumor Response and Pneumonitis in Non-Small Cell Lung Cancer Patients Treated with Immunotherapy"

_jcm, 2025, doi:10.3390/jcm14124330_

Round 1
Reviewer 1 Report
Comments and Suggestions for Authors
This study explores the use of radiomic models to predict tumor response and immunotherapy-related pneumonitis in NSCLC patients. The use of AI for feature extraction from CT scans provides a novel approach to risk stratification. The methodology is generally sound, incorporating a linear mixed-effects model for harmonization and a Random Forest algorithm for prediction. However, the study has significant limitations in terms of model performance, sample size, and generalizability, which necessitate major revisions.
- The AUC for pneumonitis prediction (0.60) and tumor response (0.63–0.66) is suboptimal for clinical application. The manuscript should address whether the model's performance is sufficient for real-world decision-making. It also mentions at r282-285 that other studies have AUC values of 0,8-0,9.
R120-121 may also be considered a limitation of the study.
- Was external validation performed? If not, the authors should acknowledge this as a limitation and discuss the need for an independent dataset.
- The dataset consists of only 159 patients, with only 31 cases of pneumonitis. This small sample size weakens the model’s ability to generalize.
- The study mentions adjusting for scanner variability but does not account for key clinical factors such as comorbidities, prior treatments, or steroid use. These should be explored or explicitly mentioned as limitations. Also confounding factors such as use of chemo in almost half of patients (r189) may alter OS results.
- The radiomics harmonization method should be further explained—how were normal spleen and lung regions selected, and how does this improve model robustness?
- It remains unclear how radiation vs. immunotherapy-related pneumonitis was distinguished. Given the challenge of differentiating the two, what measures were taken to ensure accurate classification? The manuscript only states „based on clinical judgement in medical records” (r204).
- I suggest you expand the Results section with more explanation based on the figures and tables.
- No conclusions section?
- Several grammar and readability issues:
- r32 grammatical error (a challenging issue instead of „challenging issues”)
- r43-44 the sentence must be rephrased, it makes no sense: „Patients with pneumonitis 43 development had more male, less adenocarcinoma...”
- all references should appear before the end of the sentence – before the dot.
- r96-97 please rephrase. The subject is „medical records” not „patient”. And the receive imunotherapy, not „have immunotherapy”.
- r104 should be „(5) incomplete clinical” instead of „(4)”
- The study mentions Institutional Review Board approval, but was patient consent required for retrospective analysis?
Please highlight all the limitations of the study and clearly point them out in the Discussion section. This is an example of study that clearly highlights the limitations in the Discussion section. You could also add a Figure similar to Figure 1 in this study for the selection process of your cohort.
https://doi.org/10.3390/diagnostics13101788
All in all, this study presents a valuable application of radiomics in NSCLC, but with a rather low predictive power, small sample size, and lack of external validation. If the authors can address these limitations and refine the text according to the recommendations, this study has the potential.
Comments on the Quality of English Language(Point 9 in the review report)
Please improve readability by rephrasing the sentences that do not meet an academic standard.
Reviewer 2 Report
Comments and Suggestions for Authors
Please see my comments below:
1. The AUC values for CIP prediction (0.60) and tumor response (0.63-0.66) are relatively low, suggesting moderate predictive ability. Compared to previous studies reporting higher AUC values (~0.80-0.90), the model’s clinical applicability remains questionable. I would suggest exploring alternative AI models (e.g., deep learning, ensemble models) or feature engineering techniques to improve performance.
2. I would recommend adding a more detailed breakdown in the discussion section about why the model might have lower accuracy and how future work can address these issues.
3. The model solely relies on radiomics features without integrating key clinical and molecular biomarkers such as PD-L1 expression, genomic alterations (e.g., EGFR, KRAS, TP53 mutations), and inflammatory markers (CRP, IL-6, IL-17). I would suggest incorporating radiomics + clinical + molecular data to enhance predictive accuracy.
Round 2
Reviewer 1 Report
Comments and Suggestions for Authors
The authors have modified the manuscript according to some of my suggestions and answered most of my questions.
Although the results are not spectacular in terms of clinical application, the research topic is of high interest.
Reviewer 2 Report
Comments and Suggestions for Authors
I recommend the acceptance of the current version.